# Understanding Dataset Distillation via Spectral Filtering

**Deyu Bo**[1]   **Songhua Liu**[2,1]   **Xinchao Wang**[1]*
[1]National University of Singapore    [2]Shanghai Jiao Tong University
deyu.bo@nus.edu.sg, liusonghua@sjtu.edu.cn, xinchao@nus.edu.sg

## Abstract

Dataset distillation (DD) has emerged as a promising approach to compress datasets and speed up model training. However, the underlying connections among various DD methods remain largely unexplored. In this paper, we introduce UniDD, a spectral filtering framework that unifies diverse DD objectives. UniDD interprets each DD objective as a specific filter function applied to the eigenvalues of the feature-feature correlation (FFC) matrix to extract certain frequency information of the feature-label correlation (FLC) matrix. In this way, UniDD reveals that the essence of DD fundamentally lies in matching frequency-specific features. Moreover, we characterize the roles of different filters. For example, low-pass filters, *e.g.*, DM and DC, capture blurred patches, while high-pass filters, *e.g.*, MTT and FrePo, prefer to synthesize fine-grained textures and have better diversity. However, existing methods can only learn the sole frequency information as they rely on fixed filter functions throughout distillation. To address this limitation, we further propose Curriculum Frequency Matching (CFM), which gradually adjusts the filter parameter to cover both low- and high-frequency information of the FFC and FLC matrices. Extensive experiments on small-scale datasets, such as CIFAR-10/100, and large-scale ImageNet-1K, demonstrate the superior performance of CFM over existing baselines and validate the practicality of UniDD.

## 1 Introduction

The exponential growth of data presents significant challenges to the efficiency and scalability of training deep neural networks. Dataset distillation (DD) has emerged as a solution to these challenges, aiming to condense large-scale real datasets into compact, synthetic ones without compromising model performance (Wang et al., 2018; Yu et al., 2024; Lei & Tao, 2024; Geng et al., 2023; Sachdeva & McAuley, 2023). This approach has shown promise across a range of domains, including image (Zhao et al., 2021; Zhao & Bilen, 2023; Cazenavette et al., 2022), time series (Liu et al., 2024b; Ding et al., 2024), and graph (Jin et al., 2022; Yang et al., 2023; Liu et al., 2024a).

Existing DD methods vary in their optimization objectives, which can be grouped into four main categories. Statistical matching (Zhao & Bilen, 2023; Zhao et al., 2023) aligns key statistics, such as mean and variance, between real and synthetic datasets. Gradient matching (Zhao et al., 2021; Kim et al., 2022; Zhao & Bilen, 2021) minimizes the direction of model gradients across real and synthetic data during training. Trajectory matching (Cazenavette et al., 2022; Guo et al., 2024) enforces the synthetic data to emulate the update trajectories of real model parameters. Kernel-based approaches (Zhou et al., 2022; Loo et al., 2022) employ a closed-form solution to bypass the inner optimization and improve distillation efficiency. Despite differences in their objectives, all methods strive to minimize the discrepancy between real and synthetic datasets from certain perspectives. This raises some key questions: *Are these DD methods related? If so, is there a unified framework that can encompass and explain the objectives of various DD methods?* These questions are crucial as a unified framework can deepen our understanding of DD, uncover the essence of existing methods, and offer potential insights for advanced distillation technologies.

As the first contribution of our work, we theoretically analyze some representative DD methods, including DM (Zhao & Bilen, 2023), DC (Zhao et al., 2021), MTT (Cazenavette et al., 2022), and

---

*Corresponding Author

FrePo (Zhou et al., 2022), and summarize them into a spectral filtering framework, termed UniDD, uncovering their commonalities and differences. Specifically, these DD methods are all proven to match the feature-feature correlation (FFC) and feature-label correlation (FLC) matrices between real and synthetic datasets. Their difference lies in the filter function applied to the FFC matrix, which changes the eigenvalues of the FFC matrix to extract certain frequency components of the FLC matrix. Based on their filtering behaviors, we classify existing DD methods into low-frequency matching (LFM) and high-frequency matching (HFM). Our experimental investigations, detailed in Section 4, demonstrate that LFM-based methods, *e.g.*, DM and DC, tend to learn coarse-grained colors and blur the synthetic images. In contrast, HFM-based methods, *e.g.*, MTT and FrePo, focus on fine-grained textures, which have better diversity.

The connections between DD objectives and filter functions further enable us to identify the weaknesses of existing methods. Traditional DD methods typically select the matching objectives based on intuition, while UniDD shifts this paradigm to filter design, making the new DD methods more interpretable and reliable. Based on the guidance of UniDD, we find that existing DD methods only use fixed filter functions during distillation and can only learn a single frequency component, which limits their adaptability. Therefore, we propose Curriculum Frequency Matching (CFM), which gradually adjusts the filter parameter to increase the ratio of high-frequency information in the synthetic data, thereby covering a wider range of frequency components. To verify its effectiveness, we conduct extensive experiments on CIFAR-10/100, Tiny-ImageNet, and ImageNet-1k. The results demonstrate that CFM consistently outperforms baselines by substantial margins and yields better cross-architecture generalization ability. The contributions of this paper are summarized below:

- We introduce UniDD, a spectral filtering framework that interprets each DD objective as a specific filter function applied to the FFC and FLC matrices, thus unifying diverse DD methods as a frequency-matching problem.
- We classify existing methods into low-frequency and high-frequency matching based on their filter functions, highlighting their respective roles in encoding global colors and local textures.
- We propose CFM, a novel DD method with dynamic filters that encode both low- and high-frequency information. Extensive experiments across diverse benchmarks demonstrate its superior performance over existing baselines.

## 2 PRELIMINARY

**Notations** Let $\mathcal{T} = (H, Y)$ denote a real dataset, where $H$ represents the original data with $|H| = n$ samples, and $Y \in \mathbb{R}^{n \times c}$ is the one-hot label matrix for $c$ classes. The goal of DD is to learn a synthetic network $\mathcal{S} = (H_s, Y_s)$, where $|H_s| = m \ll n$ and $Y_s \in \mathbb{R}^{m \times c}$, such that models trained on $\mathcal{T}$ and $\mathcal{S}$ have comparable performance. In addition, there is a pre-trained distillation network $\phi(\cdot)$, such as ConvNet (Zhao et al., 2021) or ResNet-18 (He et al., 2016) for the image datasets. We use $X = \phi(H) \in \mathbb{R}^{n \times d}$ and $X_s = \phi(H_s) \in \mathbb{R}^{m \times d}$ to indicate the data representations learned by the distillation network on real and synthetic datasets, where $d$ is the dimension.

**FFC and FLC matrices.** For the real and synthetic datasets, the FFC matrices are defined as $X^\top X, X_s^\top X_s \in \mathbb{R}^{d \times d}$, respectively. These matrices capture correlations among feature channels and serve as important statistical descriptors of the datasets. Similarly, the FLC matrices are denoted as $X^\top Y, X_s^\top Y_s \in \mathbb{R}^{d \times c}$, respectively. Each column corresponds to the average representation of a class, capturing the alignment between channels and labels.

**Spectral Filtering.** As $X^\top X$ is positive semi-definite, it can be decomposed into $X^\top X = U \Lambda U^\top$, where $U$ is the eigenvectors and $\Lambda$ is the diagonal eigenvalue matrix with $\lambda_i = \Lambda_{ii}$. The filtering function $f(\cdot)$ acting on $X^\top X$ is equivalent to acting on its eigenvalues, *i.e.*, $f(X^\top X) = U f(\Lambda) U^\top$. Without loss of generality, we assume that the eigenvalues are sorted in ascending order, *i.e.*, $0 \leq \lambda_1 \leq \cdots \leq \lambda_d$. Jolliffe (2011) points out that *eigenvectors associated with large eigenvalues encode low-frequency information, while small eigenvalues correspond to high-frequency components*. Therefore, a high-pass filter can be defined as $f(\lambda_i) \geq f(\lambda_{i+1}), \forall i \in [1, d-1]$, as it will assign larger amplitudes to high-frequency eigenvectors, while the low-pass filters will do the opposite.

**Spectral-domain Dataset Distillation.** There are some DD methods operating in the spectral domain, such as FreD (Shin et al., 2023) and NSD (Yang et al., 2024a). UniDD differs from them in two key aspects: First, FreD and NSD analyze the frequency of the input data itself, *e.g.*, images in the

Table 1: An overview of the objective and filter functions of four representative DD methods.

| Matching | Method | Objective Function | Filtering Function |
|---|---|---|---|
| Low-frequency | DM | $\left\|X^\top Y - X_s^\top Y_s\right\|_F^2$ | $f(\lambda) = 1$ |
| | DC | $\left\|X^\top X - X_s^\top X_s\right\|_F^2 + \left\|X^\top Y - X_s^\top Y_s\right\|_F^2$ | $f(\lambda) = \{1, \lambda\}$ |
| High-frequency | MTT | $\left\|(I - \alpha X^\top X)^P - (I - \alpha X_s^\top X_s)^Q\right\|_F^2 +$ $\alpha\left\|\sum_{p=0}^{P-1}(I - \alpha X^\top X)^p X^\top Y - \sum_{q=0}^{Q-1}(I - \alpha X_s^\top X_s)^q X_s^\top Y_s\right\|_F^2$ | $f(\lambda) = (1 - \alpha\lambda)^{\{p,q\}}$ |
| | FRePo | $\left\|(X^\top X + \beta I)^{-1} X^\top Y - (X_s^\top X_s + \beta I)^{-1} X_s^\top Y_s\right\|_F^2$ | $f(\lambda) = (\lambda + \beta)^{-1}$ |

Fourier domain, while UniDD operates on the spectral properties of the FFC matrix, making it more general and data-agnostic. Second, FreD and NSD remain data-parameterization methods whose optimization relies on existing objectives, while UniDD formulates a framework of DD objectives in the representation space, enabling the principled design of new distillation loss functions.

# 3 SPECTRAL FILTERING OF DATASET DISTILLATION

Before delving into the details, we first formally formulate UniDD, a spectral filtering framework that unifies diverse DD objectives.

**Theorem 1.** *There exists a wide range of dataset distillation methods that can be unified into a spectral filtering framework, defined as:*

$$\min_{X_s}\left\|f(X^\top X)g(X^\top Y) - f(X_s^\top X_s)g(X_s^\top Y_s)\right\|_F^2, \tag{1}$$

*where the FFC and FLC matrices serve as a filter and signal, respectively, $f(\cdot)$ is the filtering function, and $g(\cdot)$ is a binary function with $g(X^\top Y) = I$ or $X^\top Y$.*

We analyze four representative DD methods in the following and explain their relationship with different filter functions. Based on the filter behaviors, they can be divided into LFM-based and HFM-based methods. See Table 1 for a quick overview and Appendix B for more detailed derivations.

## 3.1 LOW-FREQUENCY MATCHING

Methods belonging to LFM tend to capture the principal components of the FFC matrix. Generally, these methods have a quick convergent rate and perform well with lower synthetic budgets. However, they fail to learn fine-grained information and have poor diversity. Here, we analyze two sub-categories: statistical matching and gradient matching.

**Statistical Matching.** Matching important statistics between the real and synthetic datasets is a straightforward way to distill knowledge. DM (Zhao & Bilen, 2023) proposes to match the average representations of each class, whose objective function can be defined as:

$$\left\|X^\top Y - X_s^\top Y_s\right\|_F^2, \tag{2}$$

where the corresponding filtering functions are $f(X^\top X) = I$ and $g(X^\top Y) = X^\top Y$, respectively. For clarity, we omit the mean normalization here, but this does not affect the definitions of $f$ and $g$.

However, DM does not perform well on complex distillation backbones, *e.g.*, ResNet-18, as it only matches the statistic of the last layer of backbones. To overcome this limitation, SRe$^2$L (Yin et al., 2023) proposes to match the mean and variance information in each Batch Normalization (BN) layer. The objective function is defined as:

$$\left\|\mathrm{diag}(X^\top X) - \mathrm{diag}(X_s^\top X_s)\right\|_F^2 + \left\|\mathrm{avg}(X) - \mathrm{avg}(X_s)\right\|_F^2, \tag{3}$$

where $\mathrm{diag}(\cdot)$ and $\mathrm{avg}(\cdot)$ indicate diagonal and average operations. Notably, SRe$^2$L replaces the class representation $X^\top Y$ with average sample representation $\mathrm{avg}(X)$, thus losing the category information. It then adds a cross-entropy classification loss, *i.e.*, $\mathcal{L}_{ce}(H_s, Y_s)$ as compensation.

**Gradient Matching.** Another LFM example is gradient matching (Zhao et al., 2021), which minimizes the differences of model gradients in the real and synthetic datasets. The gradients of the linear classifier are calculated as:

$$\nabla_W = X^\top(XW - Y), \ \nabla_W^s = X_s^\top(X_sW - Y_s). \tag{4}$$

By deriving the upper bound of minimizing the gradient differences, we unify gradient matching into our framework:

$$\|\nabla_W - \nabla_W^s\|_F^2 \le \|W\|_F^2\|X^\top X - X_s^\top X_s\|_F^2 + \|X^\top Y - X_s^\top Y_s\|_F^2, \tag{5}$$

where the corresponding filtering functions are $f(X^\top X) = X^\top X$ when $g(X^\top Y) = I$, and $f(X^\top X) = I$ when $g(X^\top Y) = X^\top Y$. In addition, Deng et al. (2024) adds a covariance matching loss to DM to explore the inter-feature correlations, which can also be viewed as a special case of gradient matching.

## 3.2 High-frequency Matching

Instead of directly matching the principle components, HFM-based methods typically apply high-pass filters on the FFC matrix to enhance the high-frequency information and improve diversity. They usually perform better than LFM-based methods but also bring more computations. We analyze two sub-categories: trajectory matching and kernel ridge regression (KRR).

**Trajectory Matching.** Matching the long-range training trajectories has been proven to be an effective approach for DD. We take MTT (Cazenavette et al., 2022) as an example, which minimizes the differences between network parameters trained on the real and synthetic datasets. The objective function can be formulated as:

$$\left\|W^P - W_s^Q\right\|_F^2, \tag{6}$$

where $W^P$ and $W_s^Q$ indicate the parameters trained on the real and synthetic datasets for $P$ and $Q$ iterations, respectively. For a full-batch gradient descent, we have:

$$W^1 = W^0 - \alpha\nabla_W = (I - \alpha X^\top X)W^0 + \alpha X^\top Y, \tag{7}$$

where $\alpha$ is the learning rate and $W^0$ indicates the initial parameters. For $K$ iterations, we have:

$$W^K = (I - \alpha X^\top X)^K W^0 + \sum_{k=0}^{K-1} \alpha(I - \alpha X^\top X)^k X^\top Y. \tag{8}$$

As $W^P$ and $W_s^Q$ have the same initialization $W^0$, the objective function can be reformulated as:

$$\left\|W^P - W_s^Q\right\|_F^2 \le \left\|(I - \alpha X^\top X)^P - (I - \alpha X_s^\top X_s)^Q\right\|_F^2 +$$
$$\alpha\left\|\sum_{p=0}^{P-1}(I - \alpha X^\top X)^p X^\top Y - \sum_{q=0}^{Q-1}(I - \alpha X_s^\top X_s)^q X_s^\top Y_s\right\|_F^2, \tag{9}$$

where the corresponding filtering functions are $f(X^\top X) = (I - \alpha X^\top X)^{P/Q}$ when $g(X^\top Y) = I$, and $f(X^\top X) = \sum(I - \alpha X^\top X)^{p/q}$ when $g(X^\top Y) = X^\top Y$.

Subsequent works improve MTT from different perspectives, such as memory overhead (Cui et al., 2023), trajectory training (Du et al., 2023), and trajectory selection (Guo et al., 2024). However, they all follow the same objective function so that they can be naturally included in our unified framework.

**KRR.** The early-stage DD algorithms usually update the synthetic data by solving a two-level optimization problem. To reduce computation and memory costs, KRR-based methods are proposed to replace the inner-optimization with a closed-form solution:

$$\left\|Y - \mathcal{K}_{ts}(\mathcal{K}_{ss} + \beta I)^{-1}Y_s\right\|_F^2, \tag{10}$$

where $\mathcal{K}_{ts} = \mathcal{K}(X, X_s) \in \mathbb{R}^{n \times m}$ and $\mathcal{K}_{ss} = \mathcal{K}(X_s, X_s) \in \mathbb{R}^{m \times m}$ are two Gram matrices. There are many choices of kernel functions. When using the linear kernel, we have:

$$\left\|Y - XX_s^\top(X_sX_s^\top + \beta I)^{-1}Y_s\right\|_F^2, \tag{11}$$

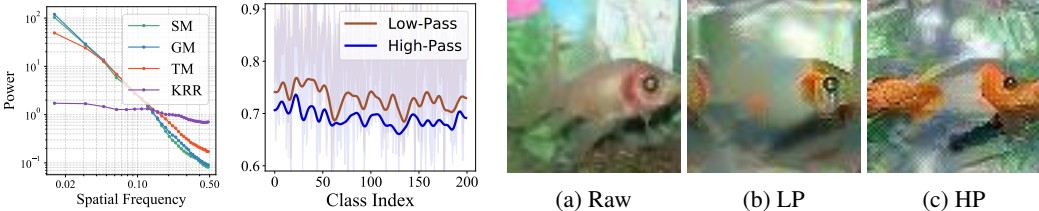

Figure 1: Power Spectral Density

Figure 2: Intra-class Similarity.

(a) Raw  (b) LP  (c) HP

Figure 3: Comparison between real and synthetic images. LP: Low-pass filter; HP: High-pass filter.

where $X_s^\top (X_s X_s^\top + \beta I)^{-1} Y_s$ can be seen as a trainable weight matrix $W_s \in \mathbb{R}^{d \times c}$. By adding a regularization term $\beta \|W_s\|_F^2$, the optimal solution of $W_s$ becomes:

$$W_s^* = (X^\top X + \beta I)^{-1} X^\top Y. \tag{12}$$

Then the objective function of KRR can be reformulated as:

$$\left\| (X^\top X + \beta I)^{-1} X^\top Y - X_s^\top (X_s X_s^\top + \beta I)^{-1} Y_s \right\|_F^2. \tag{13}$$

By applying a matrix identity transform (Welling, 2013) on the second term, the objective function can be unified into UniDD:

$$\left\| (X^\top X + \beta I)^{-1} X^\top Y - (X_s^\top X_s + \beta I)^{-1} X_s^\top Y_s \right\|_F^2, \tag{14}$$

where the corresponding filtering functions are $f(X^\top X) = (X^\top X + \beta I)^{-1}$ and $g(X^\top Y) = X^\top Y$.

It is worth noting that only the linear kernel case can be unified into our framework, *e.g.*, KIP (Nguyen et al., 2021a;b) and FrePo (Zhou et al., 2022). We leave the analysis of the non-linear kernel, *e.g.*, Gaussian and polynomial, as further work.

## 4 IN-DEPTH ANALYSIS OF FILTERS

The above analysis reveals that DD methods adopt different filters, *e.g.*, low-pass and high-pass. This section further justifies our claim and analyzes the roles of different filters through experiments on the Tiny-ImageNet dataset.

**Power Spectral Density.** We first verify that SM and GM behave as low-pass filters, whereas TM and KRR exhibit high-pass filtering. Specifically, Figure 1 visualizes the radially averaged power spectral density of the synthetic images learned by these four methods. The results show that SM and GM have more energy in the low-frequency band, while TM and KRR allocate greater power to high-frequency components, thereby supporting our claim.

**Intra-class Similarity.** We further investigate the influence of low-pass and high-pass filters on the synthetic data. Following G-VBSM (Shao et al., 2024), we compute the intra-class cosine similarity of synthetic datasets distilled by different filters. As shown in Figure 2, the low-pass filter increases intra-class similarity of synthetic data and preserves the consistency of DD, while the high-pass filter improves data diversity by reducing intra-class similarity.

**Visualization.** Finally, we visualize the raw images and the synthetic images distilled by low-pass and high-pass filters. We observe that synthetic images produced by low-pass filters (Figure 2b) appear noticeably blurrier than the original images (Figure 2a), as low-pass filters assign higher weights to eigenvectors associated with larger eigenvalues. These eigenvectors represent the low-frequency components of the original data, *e.g.*, coarse-grained shapes. In contrast, high-pass filters emphasize eigenvectors with smaller eigenvalues, thereby enhancing fine-grained and complex textures, as shown in Figure 2c.

## 5 THE PROPOSED METHOD

It is crucial to preserve both the consistency and diversity of the synthetic datasets. However, existing DD methods only have fixed filter functions and cannot capture the diverse frequency information of the real datasets, which motivates the design of our model. The framework is shown in Figure 4.

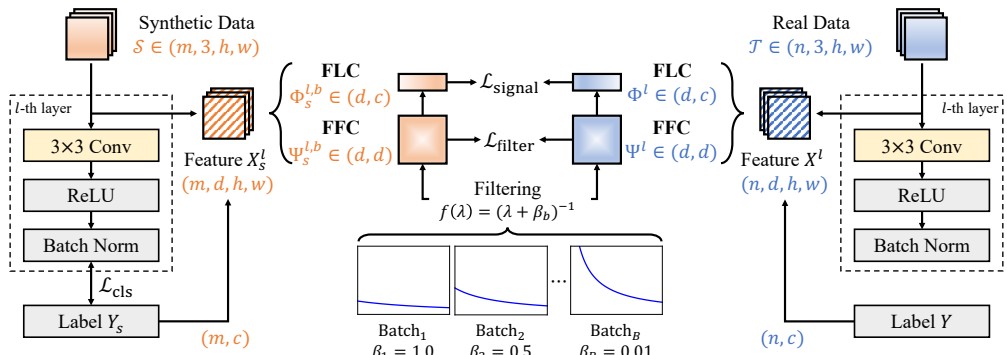

Figure 4: Framework of CFM. The real and synthetic data is first fed into a backbone to calculate their corresponding FFC and FLC matrices. The filter $f(\lambda) = (\lambda + \beta_b)^{-1}$ is then applied on FFC to extract certain frequencies. CFM uses different values of $\beta_b$ in each batch to cover both the low-frequency and high-frequency information.

**Implementation of FFC and FLC.** We calculate the FFC and FLC matrices based on the feature map of each convolutional layer. Specifically, the feature map of the $l$-th layer is denoted as $X_s^l \in \mathbb{R}^{m \times d \times h \times w}$, indicating the number of synthetic images, channels, height, and width, respectively. For the FFC matrix, we reshape the feature map into channel-level representations, and for the FLC matrix, we average it along the height and width dimensions:

$$\hat{X}_s^l = \text{reshape}(X_s^l) \in \mathbb{R}^{mhw \times d}, \quad \bar{X}_s^l = \text{avg}(X_s^l) \in \mathbb{R}^{m \times d}. \tag{15}$$

In practice, directly computing the FFC matrices suffers from numerical overflow problems because the reshape operator significantly increases the number of samples from $m$ to $mhw$. Therefore, we use the covariance matrix and mean representation as substitutes to stabilize the training process

$$\Psi_s^l = \frac{1}{mhw}(\hat{X}_s^l - \bar{X}_s^l)^\top(\hat{X}_s^l - \bar{X}_s^l), \quad \Phi_s^l = \frac{1}{m}\bar{X}_s^{l\top}Y_s, \tag{16}$$

where $\Psi_s^l$ and $\Phi_s^l$ denote the normalized FFC and FLC matrices of the synthetic dataset. Similarly, we can calculate $\Psi^l$ and $\Phi^l$ for the real dataset.

**Exponential Moving Updating (EMU).** The computation of $\Psi^l$ and $\Phi^l$ is affordable for real datasets but is much more complex for synthetic datasets. In practice, we can only calculate $\Psi_s^l$ and $\Phi_s^l$ in a mini-batch manner, which differs from the full-batch assumption in the theoretical analysis. To mitigate this gap, we adopt a moving update technology (Shao et al., 2024; Loo et al., 2024) to cache the result of each batch and ultimately approximate the statistics of the real datasets:

$$\Psi_s^{l,b} = \frac{1}{b}\Psi_s^{l,b} + (1 - \frac{1}{b})\Psi_s^{l,b-1}, \quad \Phi_s^{l,b} = \frac{1}{b}\Phi_s^{l,b} + (1 - \frac{1}{b})\Phi_s^{l,b-1}, \tag{17}$$

where $\Psi_s^{l,b}$ and $\Psi_s^{l,b-1}$ indicate the FFC matrices of the current $b$-th batch and the previous batch, respectively. The symbols of the FLC matrices are the same as above.

**Curriculum Frequency Matching.** After calculating the FFC and FLC matrices, we need to design a filtering function to match the crucial frequency information between the real and synthetic datasets. Specifically, we consider a filter $f(\lambda) = (\lambda + \beta)^{-1}$, whose shape is controlled by the value of $\beta$. Generally, as $\beta$ decreases, the proportion of high-frequency information gradually increases, thus improving the diversity of synthetic datasets. However, high-frequency information also introduces additional noise, resulting in performance degradation, as shown in Figure 6. Therefore, we define a scheduler to dynamically adjust the value of $\beta$ in different batches to balance the consistency and diversity of the synthetic dataset

$$\beta_b = \beta * (1 + \cos(\pi b/B))/2, \tag{18}$$

where $\beta$ is a hyper-parameter that controls the maximum frequency, and $B$ denotes the total number of batches. Notably, $f(\Psi_s^{l,b}) = (\Psi_s^{l,b} + \beta_b)^{-1}$ has different characteristics. For example, $\Psi_s^{l,1}$, corresponding to $\beta_1$, preserves consistency and $\Psi_s^{l,B}$, corresponding to $\beta_B$, emphasizes diversity.

Table 2: Performance (%) of different DD methods in CIFAR-10/100. The best performance is highlighted in **bold**. Results are taken from the original papers, and $-$ indicates missing data.

| Dataset | IPC | ConvNet-128 | | | | | | | ResNet-18 | | | |
|---------|-----|------|------|------|------|--------|------|------|---------------|--------|------|------|
| | | DM | DC | MTT | FrePo | G-VBSM | DWA | CFM | SRe$^2$L | G-VBDM | DWA | CFM |
| CIFAR-10 | 10 | 48.9±0.6 | 44.9±0.5 | 65.3±0.7 | **65.5±0.4** | 46.5±0.7 | 45.0±0.4 | 52.1±0.5 | 27.2±0.5 | 53.5±0.6 | 32.6±0.4 | **57.0±0.3** |
| | 50 | 63.0±0.4 | 53.9±0.5 | 71.6±0.2 | **71.7±0.2** | 54.3±0.3 | 63.3±0.7 | 64.0±0.4 | 47.5±0.6 | 59.2±0.4 | 53.1±0.3 | **82.3±0.4** |
| CIFAR-100 | 10 | 29.7±0.3 | 25.2±0.3 | 33.1±0.4 | 42.5±0.2 | 38.7±0.2 | 47.6±0.4 | **58.3±0.4** | 31.6±0.5 | 59.5±0.4 | 39.6±0.6 | **64.6±0.4** |
| | 50 | 43.6±0.4 | 30.6±0.6 | 42.9±0.3 | 44.3±0.2 | 45.7±0.4 | 59.0±0.1 | **67.1±0.3** | 49.5±0.3 | 65.0±0.5 | 60.3±0.5 | **71.4±0.2** |

Table 3: Performance (%) of different DD methods in Tiny-ImageNet and ImageNet-1k.

| Dataset | IPC | ResNet-18 | | | | | | ResNet-101 | | | |
|---------|-----|---------|------|--------|---------|------|------|---------|---------|------|------|
| | | SRe$^2$L | CDA | G-VBSM | RDED | DWA | CFM | SRe$^2$L | RDED | DWA | CFM |
| T-ImageNet | 50 | 41.4±0.4 | 48.8 | 47.6±0.3 | **58.2±0.1** | 52.8±0.2 | 58.0±0.2 | 42.5±0.2 | 41.2±0.1 | 54.7±0.3 | **60.4±0.2** |
| | 100 | 49.7±0.3 | 53.2 | 51.0±0.4 | $-$ | 56.0±0.2 | **59.2±0.1** | 51.5±0.3 | $-$ | 57.4±0.3 | **61.1±0.2** |
| ImageNet-1k | 10 | 21.3±0.6 | $-$ | 31.4±0.5 | **42.0±0.1** | 37.9±0.2 | 40.6±0.3 | 30.9±0.1 | **48.3±1.0** | 46.9±1.4 | 47.2±0.5 |
| | 50 | 46.8±0.2 | 53.5 | 51.8±0.4 | 56.5±0.1 | 55.2±0.2 | **57.3±0.2** | 60.8±0.5 | 61.2±0.4 | 63.3±0.7 | **63.5±0.2** |
| | 100 | 52.8±0.3 | 58.0 | 55.7±0.4 | $-$ | 59.2±0.3 | **59.5±0.2** | 62.8±0.2 | $-$ | 66.7±0.2 | **66.9±0.3** |

**Loss Function.** After defining the filtering function $f$, another consideration is the choice of $g$, which is restricted to $g(X^\top Y) = I$ or $X^\top Y$. The former corresponds to a filter-matching loss, while the latter leads to a signal-matching loss. In the context of CFM, we define the loss functions as:

$$\mathcal{L}_{\text{filter}} = \sum_{b=1}^{B} \sum_{l=1}^{L} \left\| (\Psi^l + \beta_b I)^{-1} - (\Psi_s^{l,b} + \beta_b I)^{-1} \right\|,$$

$$\mathcal{L}_{\text{signal}} = \sum_{b=1}^{B} \sum_{l=1}^{L} \left\| (\Psi^l + \beta_b I)^{-1} \Phi^l - (\Psi_s^{l,b} + \beta_b I)^{-1} \Phi_s^l \right\|, \tag{19}$$

By combining these two functions with the basic classification loss, we get the final loss function:

$$\mathcal{L} = \mathcal{L}_{\text{cls}}(H_s, Y_s) + \eta \mathcal{L}_{\text{filter}} + \eta \mathcal{L}_{\text{signal}} \tag{20}$$

where $\eta = 0.1$ is a hyperparameter for all datasets. See Appendix C for more details, such as the pseudo algorithm.

## 6 EXPERIMENTS

### 6.1 EXPERIMENTAL SETUP

**Datasets.** We consider four datasets, including CIFAR-10/100 (Krizhevsky et al., 2009) (32×32, 10/100 classes), Tiny-ImageNet (Le & Yang, 2015) (64×64, 200 classes), and ImageNet-1K (Deng et al., 2009) (224×224, 1000 classes).

**Network Architectures.** We use ResNet-18 (He et al., 2016) as the distillation network for all datasets. For a fair comparison with traditional DD methods, we also use ConvNet-128 (Zhao et al., 2021) in CIFAR-10/100, as suggested by G-VBSM (Shao et al., 2024). We adopt various memory budgets for different datasets, including images per class (IPC)-10, 50, and 100.

**Baselines.** Traditional DD methods work on small-scale datasets, *e.g.*, CIFAR-10/100, but do not perform well in the large-scale ImageNet-1K. In this case, we select DM (Zhao & Bilen, 2023), DC (Zhao et al., 2021), MTT (Cazenavette et al., 2022), and FrePo (Zhou et al., 2022) as baselines in CIFAR-10/100. Moreover, we report the performance of some more recent DD methods, including SRe$^2$L (Yin et al., 2023), CDA (Yin & Shen, 2023), G-VBSM (Shao et al., 2024), RDED (Sun et al., 2024), and DWA (Du et al., 2024).

**Evaluation.** In the evaluation phase, we adopt the Fast Knowledge Distillation (FKD) strategy suggested by SRe$^2$L, which has been proven to be useful for large-scale datasets. For a fair comparison, we strictly control the hyperparameters of FKD to be the same as those of previous methods. See Appendix C.3 for more details.

Table 4: Cross-architecture performance (%) of different DD methods. ‡ indicates that the results are our evaluation; otherwise, we adopt the results from the original paper. The results of DWA are not reported due to the lack of source code and synthetic images.

| ImageNet-1k (IPC=50) | Validaton Model | | | | | |
|---|---|---|---|---|---|---|
| | ResNet-18 | ResNet-50 | ResNet-101 | DenseNet-121 | RegNet-Y-8GF | ConvNeXt-Tiny |
| SRe$^2$L | 46.80 | 55.60 | 60.81 | 49.47 | 60.34 | 53.53 |
| CDA | 53.45 | 61.26 | 61.57 | 57.35 | 63.22 | 62.58 |
| G-VBSM‡ | 51.8 | 58.7 | 61.0 | 58.47 | 62.02 | 61.93 |
| CFM | **57.32** | **63.23** | **63.46** | **60.91** | **64.03** | **64.89** |

Table 5: Ablation studies on the loss functions of CFM. Experiments are conducted on three datasets with IPC=50.

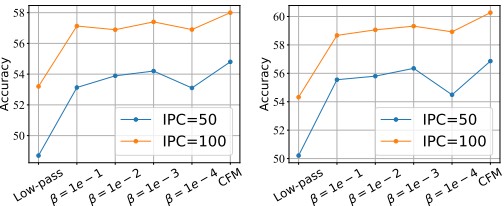

| Loss Functions | | | Datasets | | |
|---|---|---|---|---|---|
| $\mathcal{L}_{\text{filter}}$ | $\mathcal{L}_{\text{signal}}$ | $\mathcal{L}_{\text{cls}}$ | CIFAR | Tiny | ImageNet |
| ✓ | | | 64.95 | 51.20 | 53.43 |
| ✓ | ✓ | | 67.32 | 51.92 | 55.76 |
| ✓ | ✓ | ✓ | 72.48 | 54.84 | 57.32 |

Table 6: Ablation studies on filters. Left: Tiny-ImageNet; Right: ImageNet-1K.

## 6.2 QUANTITATIVE RESULTS

We conduct experiments on both small-scale and large-scale datasets. The results are shown in Tables 2 and 3, respectively, from which we have the following observations:

**CIFAR-10/100.** Traditional DD methods perform well on datasets with fewer classes, *i.e.*, CIFAR-10, and shallow models, *i.e.*, ConvNet-128. The advanced DD methods are more effective when the dataset becomes larger and the network goes deeper. We can observe that CFM consistently outperforms all advanced DD methods by a large margin. Notably, ResNet-18 trained on the original datasets has an accuracy of 94.25% and 77.45% on CIFAR-10/100. The performance of existing methods is far from achieving optimal results, while CFM is significantly approaching, demonstrating its effectiveness in distilling low-resolution data.

**Tiny-ImageNet & ImageNet-1k.** In the large-scale datasets, we use ResNet-{18, 101} to evaluate the performance of different methods. In Tiny-ImageNet, CFM outperforms DWA by 3% on average and is slightly lower than RDED when IPC=50. When IPC increases to 100, CFM achieves the best performance, demonstrating its effectiveness. In the ImageNet-1k part, we can see that CFM consistently surpasses all training-based DD methods in IPC={50, 100}, validating its superiority over the advanced DD methods. Moreover, we observe that RDED performs better than CFM in IPC=10. We suspect that the synthetic data cannot effectively cover the useful frequency features under the limited memory budget, leading to the performance degeneration of CFM.

## 6.3 CROSS-ARCHITECTURE GENERALIZATION

In addition to classification accuracy, cross-architecture generalization is also crucial for DD. Ideally, synthetic datasets should encode the knowledge of real datasets rather than overfitting to a specific model architecture. Therefore, we use diverse networks, including ResNet-50/101, DenseNet-121 (Huang et al., 2017), RegNet-Y-8GF (Xu et al., 2023), and ConvNeXt-Tiny (Liu et al., 2022b), to evaluate the generalization ability of different DD methods on the ImageNet-1k dataset with IPC=50. The distillation network is uniformly set to ResNet-18. Results are shown in Table 4, from which we can see that CFM achieves the best performance by a substantial margin across different architectures.

## 6.4 ABLATION STUDIES

We conducted two experiments to demonstrate the importance of each module in the proposed model. The first experiment, shown in Table 5, is used to identify the effectiveness of different loss functions.

Table 7: Time overhead of three DD methods.     Table 8: Space overhead of three DD methods.

| Time (s) | $\mathcal{L}_{cls}$ | $+\mathcal{L}_{BN}$ | $+\mathcal{L}_{dense}$ | $+\mathcal{L}_{filter} + \mathcal{L}_{signal}$ |
|---|---|---|---|---|
| SRe$^2$L | 0.16 | **0.26** | - | - |
| G-VBSM | 0.16 | - | 0.35 | - |
| CFM | 0.16 | - | - | 0.31 |

| Space (MB) | $\mathcal{L}_{cls}$ | $+\mathcal{L}_{BN}$ | $+\mathcal{L}_{dense}$ | $+\mathcal{L}_{filter} + \mathcal{L}_{signal}$ |
|---|---|---|---|---|
| SRe$^2$L | 17186 | 22522 | - | - |
| G-VBSM | 17186 | - | 23170 | - |
| CFM | 17186 | - | - | **21450** |

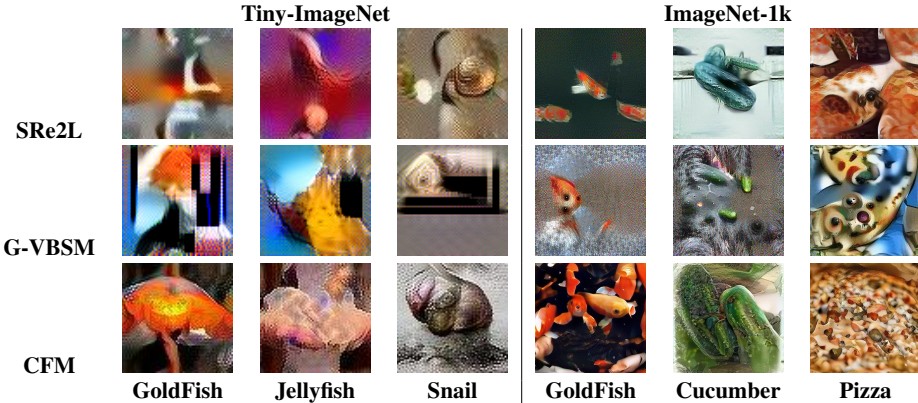

Figure 5: Visualization of the images synthesized by different DD methods.

Specifically, we sequentially remove the classification loss and signal-matching loss, and evaluate CFM on three datasets, including CIFAR-100, Tiny-ImageNet, and ImageNet-1k. We can observe that all three loss functions contribute to the performance of CFM, and the filter-matching loss plays a fundamental role. This discovery suggests that important knowledge of the real datasets may be encoded in the FFC matrix of the feature maps.

The second experiment is used to verify the effectiveness of the curriculum strategy. Specifically, we evaluate the performance of different filters, including low-pass filter, high-pass filter with parameter $\beta = \{1e^{-1}, e^{-2}, 1e^{-3}, 1e^{-4}\}$, and high-pass filter with CFM. The results are shown in Figure 6, from which we can find that the performance of the low-pass filter is far away from the high-pass filters, indicating that the high-frequency information is more important for DD. However, the extremely high frequency will also lead to performance degradation. For example, the performance of $\beta = 1e^{-4}$ is not as good as other parameters. On the other hand, CFM balances the ratio between low-frequency and high-frequency by setting a proper frequency band, consistently outperforming other filters.

## 6.5 TIME AND SPACE OVERHEAD

We compare the time and space overhead of CFM with two representative DD methods: SRe2L Yin et al. (2023) and G-VBSM Shao et al. (2024) in the ImageNet-1k dataset with a ResNet-18 backbone. To ensure fairness, all methods optimize 500 synthetic images per step. The results are averaged over 1,000 iterations on a RTX 4090 GPU. Results are shown in Tables 7 and 8.

All methods use the basic classification loss. SRe2L proposes a batch normalization loss, $\mathcal{L}_{BN}$, to align the mean and variance statistics. Building on this, G-VBSM further adds a densification loss, $\mathcal{L}_{dense}$, to improve the diversity of the synthesized images. Unlike them, CFM uses the filter-matching and signal-matching losses. Compared to SRe2L, CFM has a slightly higher time overhead, but it is still lower than G-VBSM. On the other hand, CFM has the lowest space overhead, as the number of convolutional layers is less than that of batch normalization, demonstrating the efficiency of CFM.

## 6.6 VISUALIZATION

To provide a more intuitive comparison of various DD methods, we visualize the synthetic images generated by SRe$^2$L, G-VBSM, and CFM in Figure 5. It can be observed that SRe$^2$L and G-VBSM struggle to produce meaningful images, particularly with the low-resolution Tiny-ImageNet dataset. In contrast, the images synthesized by CFM exhibit clear semantic information, *e.g.*, recognizable

shapes, demonstrating the effectiveness of matching frequency-specific features between the real and synthetic datasets. See Appendix D for more visualizations.

## 7 RELATED WORK

**Data Parameterization.** Traditionally, the synthetic images are optimized in the pixel space, which is inefficient due to the potential data redundancy. Therefore, some methods explore how to parameterize the synthetic images. IDC (Kim et al., 2022) distillates low-resolution images to save budget and up-samples them in the evaluation stage. Haba (Liu et al., 2022a) designs a decoder network to combine different latent codes for diverse synthetic datasets. RDED (Sun et al., 2024) stitches the core patches of real images together as the synthetic data. Some methods use generative models to decode the synthetic datasets from latent codes, including GAN-based methods (Zhao & Bilen, 2022; Liu & Wang, 2023), diffusion-based methods (Gu et al., 2024; Su et al., 2024), and implicit function-based methods (Shin et al., 2023). Generally, data parameterization can be incorporated with different DD objectives to improve their efficiency.

**Model Augmentation.** The knowledge of the datasets is mostly encoded in the pre-trained distillation networks. Therefore, some methods focus on training a suitable network for DD by designing some augmentation strategies. FTD (Du et al., 2023) constrains model weights to achieve a flat trajectory and reduces the accumulated errors. Zhang et al. (2023) use early-stage models and parameter perturbation to increase the search space of the parameters. More recently, DWA (Du et al., 2024) employs directed weight perturbations on the pre-training model that maintain the unique features of each synthetic data.

**Theory.** Some works tend to explore DD from a theoretical perspective. For example, Maalouf et al. (2023) analyzes the size and approximation error of the synthetic datasets. Cui et al. (2024) explores the influence of spurious correlations in DD. Yang et al. (2024b) explains the information captured by the synthetic datasets. Kungurtsev et al. (2024) provides a formal definition of DD and gives a foundation analysis on the optimization of DD. These methods have some important insights, but fail to explain the roles of existing methods. In contrast, UniDD establishes the relationship between DD objectives and spectral filtering.

## 8 CONCLUSION

In this paper, we introduce UniDD, a framework that unifies various DD objectives through spectral filtering. UniDD demonstrates that each DD objective corresponds to a filter function applied to the FFC and FLC matrices. This finding reveals the nature of various DD methods and inspires the design of new methods. Based on UniDD, we propose CFM to encode both low- and high-frequency information by gradually changing the filter parameter. Experiments conducted on various datasets validate the effectiveness and generalization of the proposed method. A promising future direction is to generalize UniDD to the distillation of unsupervised and multi-modal datasets.

## ACKNOWLEDGMENT

This project is supported by the Ministry of Education, Singapore, under its Academic Research Fund Tier 2 (Award Number: MOE-T2EP20122-0006).

## ETHICS STATEMENT

This work aims to provide a unified view of dataset distillation and deepen our understanding of data efficiency. There are many potential societal consequences of our work, none of which we feel must be specifically highlighted here.

## REPRODUCIBILITY

To ensure reproducibility, we provide additional implementation details in Appendix C. During the review process, we submit the source code as supplementary material, and we promise to release the code publicly upon acceptance of the paper.

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

## A  STATEMENT ON LLM USAGE

In drafting this manuscript, LLMs were used exclusively for polishing language and checking grammar. The content was not directly copied from LLM outputs, and all scientific concepts, analyses, and conclusions represent the authors' original work.

## B  DERIVATION OF UNIDD

The derivation of statistical matching, gradient matching, and trajectory matching is relatively intuitive and has been fully introduced in the main text. Here, we mainly derive the conclusion of KRR, which is closely related to the proposed model. Generally, the objective of KRR is defined as:

$$\mathcal{L}_{KRR} = ||\mathcal{K}W - Y||_F^2 + \beta W^\top \mathcal{K}W, \tag{21}$$

where $\mathcal{K} \in \mathbb{R}^{n \times n}$ is the Gram matrix of input data.

By taking the derivative with respect to $W$, we have:

$$\nabla_W = \mathcal{K}^\top (\mathcal{K}W - Y) + \beta(\mathcal{K}W + \mathcal{K}^\top W). \tag{22}$$

When set the gradient to zero, we have:

$$W^* = (\mathcal{K} + \beta I)^{-1} Y. \tag{23}$$

This equation gives an optimal solution of $W$, which can be used to replace the inner loop of DD. If implemented with the linear kernel, we have:

$$W_s = (X_s X_s^\top + \beta I)^{-1} Y_s. \tag{24}$$

The outer loop of DD treats $W_s$ a classifier to minimize the classification error in the real datasets:

$$\|Y - \mathcal{K}_{ts} W_s\|^2 = \left\| Y - X X_s^\top (X_s X_s^\top + \beta I)^{-1} Y_s \right\|_F^2, \tag{25}$$

which derivatives Equation 11 in the main text.

Another detail is the identity transformation, formulated as

$$\mathbf{P}(\mathbf{QP} + \mathbf{I})^{-1} = (\mathbf{PQ} + \mathbf{I})^{-1}\mathbf{P}. \tag{26}$$

By setting $X_s^\top = \mathbf{P}$ and $X_s = \mathbf{Q}$, we have:

$$X_s^\top (X_s X_s^\top + \beta I)^{-1} = (X_s^\top X_s + \beta I)^{-1} X_s^\top. \tag{27}$$

Based on this equation, we successfully transform the Gram matrix into the FFC matrix and unify KRR-based methods into UniDD.

## C    IMPLEMENTATION DETAILS.

### C.1    ALGORITHM

Algorithm 1 illustrates the distillation process of CFM.

---

**Algorithm 1** Curriculum Frequency Matching

---

**Input:** Distillation network $\phi$, real dataset $\mathcal{T} = (H, Y)$, minimum parameter $\beta_{\min}$, maximum step $T$, number of iteration $\mathcal{I}$, batch size $|B|$.
**Output:** Synthetic dataset $\mathcal{S} = (H_s, Y_s)$
 1: Feed $H$ into $\phi$ for a forward pass
 2: **for** layer index $l = 1, \cdots, L$ **do**
 3:      Calculate $\Psi^l$ and $\Phi^l$ based on Eq. 16
 4: **end for**
 5: **for** batch index $b = 1, \cdots, B$ **do**
 6:      Initialize $H_s^b$ with randomly sampled real data
 7:      Update $\beta_b$ based on Eq. 18
 8:      **repeat**
 9:          Feed $H_s^b$ into $\phi$ for a forward pass
10:          **for** layer index $l = 1, \cdots, L$ **do**
11:              Calculate $\Psi_s^{l,b}$ and $\Phi_s^{l,b}$ based on Eq. 17
12:              Calculate $\mathcal{L}_{\text{filter}}$ and $\mathcal{L}_{\text{signal}}$ based on Eq. 19
13:          **end for**
14:          Calculate $\mathcal{L}_{\text{ce}}$ and back-propagate $\mathcal{L}$
15:          Update $H_s^b$
16:      **until** Reached the number of iteration $\mathcal{I}$
17: **end for**

---

### C.2    ENVIRONMENT

All experiments are conducted on a single GeForce RTX 4090.

Table 9: Statistics of datasets.

|  | CIFAR-10 | CIFAR-100 | Tiny-ImageNet | ImageNet-1k |
| --- | --- | --- | --- | --- |
| Classes | 10 | 100 | 200 | 1000 |
| Training | 50,000 | 50,000 | 100,000 | 1,281,167 |
| Validation | 10,000 | 10,000 | 10,000 | 50,000 |
| Resolution | 64×64 | 64×64 | 64×64 | 224×224 |

Table 10: Squeeze Configurations of CFM.

| Config | CIFAR-10/100 | Tiny-ImageNet | ImageNet-1k |
| --- | --- | --- | --- |
| Epoch | 200 / 100 | 50 | 90 |
| Optimizer | SGD | SGD | SGD |
| Learning Rate | 0.1 | 0.2 | 0.1 |
| Momentum | 0.9 | 0.9 | 0.9 |
| WeightDecay | 5e-4 | 1e-4 | 1e-4 |
| BatchSize | 128 | 256 | 256 |
| Scheduler | Cosine | Cosine | Step (0.1 / 30 epochs) |
| Augmentation | RandomCrop HorizontalFlip | RandomResizedCrop HorizontalFlip | RandomResizedCrop HorizontalFlip |

Table 11: Recover Configurations of CFM.

| Config | CIFAR-10/100 | Tiny-ImageNet | ImageNet-1k |
| --- | --- | --- | --- |
| $\beta$ | 0.1 | 1.0 | 0.1 |
| Iteration | 1000 | 1000 | 1000 |
| Optimizer | Adam | Adam | Adam |
| Learning Rate | 0.25 | 0.1 | 0.1 |
| Betas | (0.5, 0.9) | (0.5, 0.9) | (0.5, 0.9) |
| BatchSize | 10/100 | 200 | 500 |
| Scheduler | Cosine | Cosine | Cosine |
| Augmentation | RandomResizedCrop HorizontalFlip | RandomResizedCrop HorizontalFlip | RandomResizedCrop HorizontalFlip |

Table 12: Recover & Validation Configurations of CFM.

| Config | CIFAR-10/100 | Tiny-ImageNet | ImageNet-1k |
| --- | --- | --- | --- |
| Epoch | 1000 | 300 | 300 |
| Optimizer | Adam / SGD | SGD | Adam |
| Learning Rate | 1e-3 / 1e-1 | 2e-1 | 1e-3 |
| Parameters | - / Mom=0.9 | Mom=0.9 | - |
| WeightDecay | 5e-4 | 1e-4 | 1e-4 |
| BatchSize | 64 | 64 | 100 |
| Scheduler | Cosine | Cosine | Cosine |
| Temperature | 30 | 20 | 20 |
| Augmentation | RandomCrop HorizontalFlip | RandomResizedCrop HorizontalFlip | RandomResizedCrop HorizontalFlip |

## C.3 EXPERIMENTAL SETUP

We use the squeeze, recover, and relabel pipeline, introduced by SRe$^2$L, for the distillation of CFM. The details of datasets, setup (*e.g.*, optimizer), and hyper-parameters are listed in Tables 9, 10, 11, and 12.

## D VISUALIZATION

Finally, we visualize more synthetic images of ImageNet-1k in Figure 6 to provide a comprehensive exhibition. Specifically, each column represents the images sampled from the same batch. From left to right, the value of $\beta_d$ decreases, and the high-frequency information gradually increases. It can be observed that images on the left side are somewhat blurry, while images on the right side have more complex textures, validating the conclusions in Section 4.

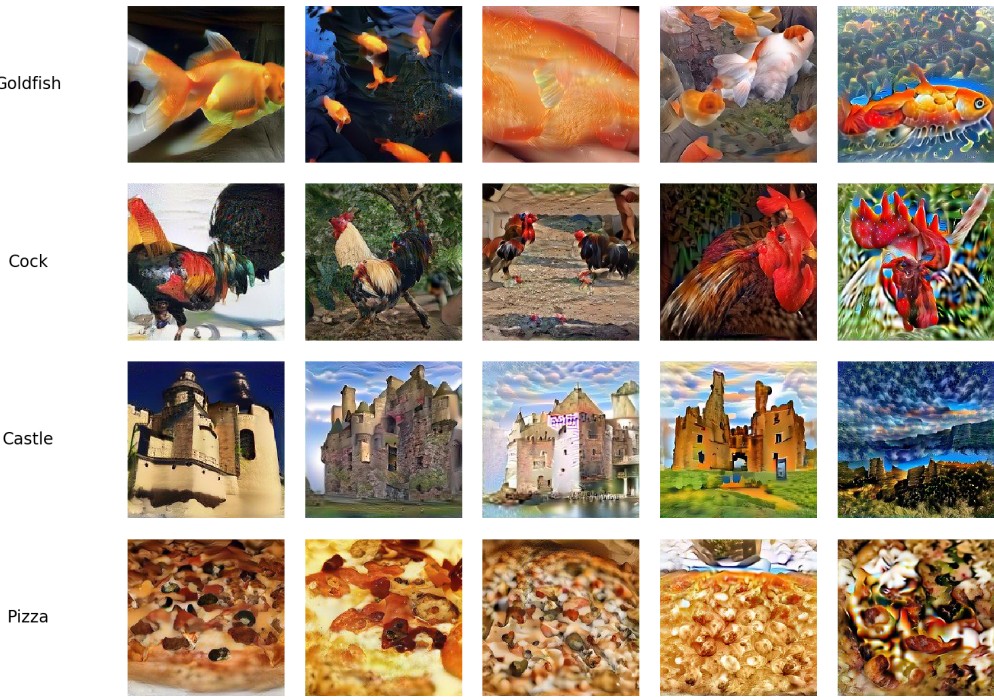

Figure 6: Synthetic images. From left to right, the high-frequency gradually increases.

