# OpenReview forum: "Understanding Dataset Distillation via Spectral Filtering"
_ICLR.cc/2026/Conference — ICLR 2026 Poster_

### Official Review · Reviewer_b6oX · 2025-10-25

**Soundness:** 2
**Presentation:** 3
**Contribution:** 2
**Rating:** 4
**Confidence:** 3

**Summary:**

This paper proposes a novel framework, UniDD, that unifies existing Dataset Distillation (DD) methods from the perspective of spectral filtering. The authors interpret different distillation objective functions as filtering functions applied to the feature-feature correlation matrix (FFC) and the feature-label correlation matrix (FLC), revealing that the essence of dataset distillation is "matching frequency-specific features." To address the limitations that existing methods use fixed filters and thus cannot simultaneously learn across multiple frequency bands, the authors further propose Curriculum Frequency Matching (CFM). This method dynamically adjusts filter parameters, gradually covering from low to high frequencies during training, thereby achieving a balance between consistency and diversity in the synthetic data.

**Strengths:**

The presentation is clear and easy to follow.

**Weaknesses:**

1.	The paper lacks sufficient rigorous theoretical justification, and UniDD's unified analysis relies on idealized assumptions. For example, the derivation of the DC method considers only linear classifiers, while the entire network is actually considered, which reduces readability and persuasiveness.
2.	CFM requires computing and matching FFC/FLC on each feature layer and applying dynamic filtering, which inevitably incurs significant computational overhead when the network is deep.

**Questions:**

1.	What are the advantages of the proposed “spectral filtering” compared to transforming an image from the spatial domain to the frequency domain?
2.	Can the authors directly demonstrate the effects of what they define as low-frequency matching and high-frequency matching, rather than only visualizing them via filtered images as in Fig. 1?
3.	Please provide a comparison of computational complexity metrics.
4.	Can the ablation study include comparisons against baseline methods? Moreover, in Table 5, the performance improvement of the designed loss over the basic classification loss seems relatively weak.
5.	In Table 2, CFM achieves very strong performance on CIFAR-100. Does it surpass training on the full real dataset (Whole Dataset), or how close does it get?

---

> ### Author Response · Authors · 2025-11-30
> **Response to Reviewer b6oX (1/2)**
>
> We sincerely appreciate your thoughtful feedback and insightful questions.
>
> &nbsp;
>
> > **W1: Theoretical justification**
>
> A1: Our theoretical framework is based on the analysis of a linear classifier. We think existing results have provided enough insights about dataset distillation. The reasons are two-fold:
>
> - Many existing methods employ fixed encoders during the distillation process. For example, FrePo [1] uses fixed image encoders and models the linear classifier as a closed-form regression loss, which has achieved considerable results. Moreover, SRe2L [2] also uses a fixed ResNet-18 as the distillation backbone and extends DD to ImageNet-1k successfully.
> - Although gradient matching and trajectory matching use full parameters, they do not show competitive results on large-scale datasets, e.g., ImageNet-1k. Moreover, some recent works [3, 4] show that matching gradients or trajectories of the last layer can obtain better distillation performance, which supports our theoretical analysis.
>
> [1] Dataset Distillation using Neural Feature Regression.
>
> [2] Squeeze, Recover and Relabel: Dataset Condensation at ImageNet Scale From A New Perspective.
>
> [3] Dataset Distillation for Pre-Trained Self-Supervised Vision Models.
>
> [4] Beyond Modality Collapse: Representations Blending for Multimodal Dataset Distillation.
>
> &nbsp;
>
> > **W2 & Q3: Complexity of CFM and computational overhead comparison**
>
> A2: CFM is computationally friendly. It achieves minimal memory allocation while only slightly increasing the time complexity over baselines.
>
> ``[Efficiency of CFM]``
>
> The computational costs of CFM are concentrated in the matrix inversion. We clarify that these operators are inexpensive for two reasons:
>
> - **Spatial pooling.** For a feature map $X \in \mathbb{R}^{N \times C \times H \times W}$, we average it along the height (H) and width (W) dimensions and preserve the channel (C) dimension only, leaving at most a $C \times C$ covariance matrix. In ResNet-18, $C_{max}=512$, and the computation of a 512*512 matrix inversion is affordable.
> - **Pre-computing statistics of real data.** To reduce complexity, the matrix inversions of real data, i.e., $(\Psi + \beta_b I)^{-1}$ and $(\Psi + \beta_b I)^{-1}\Phi$, are computed once, off‑line, and reused during distillation. Hence, they add no cost to the training loop.
>
> ``[Time and Space Overhead]``
>
> We compare the overhead of CFM with two representative baselines: SRe2L and G-VBSM in the ImageNet-1k dataset with a ResNet‑18 backbone.
>
> To ensure fairness, all methods optimize 500 synthetic images per step. The results are averaged over 1,000 iterations on a 4090 GPU.
>
> Notably, all three methods use the basic classification loss, i.e., $\mathcal{L}_{cls}$. Therefore, we first report the overhead of a single classification loss and then show the overhead of adding additional matching losses.
> - SRe2L uses the BN-matching loss, i.e., $ \mathcal{L}\_{cls} + \mathcal{L}\_{BN} $
> - G-VBSM adds a data densification loss in SRe2L, i.e., $ \mathcal{L}\_{cls} + \mathcal{L}\_{BN} + \mathcal{L}\_{dense} $
> - CFM uses the filter and signal matching losses, i.e., $ \mathcal{L}\_{cls} + \mathcal{L}\_{filter} + \mathcal{L}\_{signal}$.
>
> The results are shown below
>
> | Time (s / iter) | $\mathcal{L}_{cls}$ | $+\mathcal{L}_{BN}$ | $+ \mathcal{L}\_{BN} + \mathcal{L}\_{dense}$ | $+\mathcal{L}\_{filter} + \mathcal{L}\_{signal}$ |
> | :--: | :--: | :--: | :--: | :--: |
> | SRe2L | 0.16 | **0.26** | - | - |
> | G-VBSM | 0.16 | - | 0.35 | - |
> | CFM | 0.16 | - | - | 0.31 |
>
> | Space (MB) | $\mathcal{L}_{cls}$ | $+\mathcal{L}_{BN}$ | $+ \mathcal{L}\_{BN} + \mathcal{L}\_{dense}$ | $+\mathcal{L}\_{filter} + \mathcal{L}\_{signal}$ |
> | :--: | :--: | :--: | :--: | :--: |
> | SRe2L | 17186 | 22522 | - | - |
> | G-VBSM | 17186 | - | 23170 | - |
> | CFM | 17186 | - | - | **21450** |
>
> We have the following observations:
>
> - **In the time overhead**, SRe2L runs faster than G-VBSM and CFM, indicating that BN-matching is efficient than covariance-based matching loss. G-VBSM proposes the data densification loss, which aims to improve the rank of the covariance matrix. As it involves both decomposition and BN-matching, G-VBSM has the largest time overhead.
> - **In the space overhead**, it is surprising that CFM has the smallest space allocation. The reason is that CFM is calculated on the convolutional layer of ResNet-18, which has **17** Conv2d modules, while SRe2L relies on the batch normalization layer of ResNet-18, which has **19** BatchNorm2d modules. G-VBSM uses both conv and BN layers and has the largest space overhead.

---

> ### Author Response · Authors · 2025-11-30
> **Response to Reviewer b6oX (2/2)**
>
> > **Q1: Matrix Frequency and Image Frequency**
>
> A3: Some existing works, e.g., FreD [5], leverage the Fourier transform to convert images to the frequency domain and perform dataset distillation. While effective, these works differ from UniDD in the following perspectives.
>
> - UniDD studies the **dataset-level frequency**, i.e., $X^{T}X$, where $X \in \mathbb{R}^{n \times d}$. In contrast, FreD focuses on **sample-level frequency**, i.e., a single image, which leverages the DCT to parametrize each synthetic image.
> - UniDD mainly analyzes the **objective functions** of various DD methods and directly optimizes the synthetic images in the pixel space. On the contrary, FreD is a **data parameterization** method, which updates the synthetic images in the spectral domain. It can be combined with an arbitrary DD objective.
> - In summary, UniDD and FreD are two orthogonal research lines that can promote each other. We will cite this work and add this discussion in the revision.
>
> [5] Frequency Domain-based Dataset Distillation.
>
> &nbsp;
>
> > **Q2: Explanation of low-frequency matching and high-frequency matching**
>
> A4: The fundamental principle of frequency matching is about matching the eigenvalues and eigenvectors of the FFC and FLC matrices. As explained in lines 100-107, eigenvectors corresponding to larger eigenvalues encode low-frequency information, and vice versa.
>
> Let $\Phi=U \Lambda U^{\top}$ and $\hat{\Phi}=\hat{U}\hat{\Lambda}\hat{U}^{\top}$ denote the FFC matrix of the real and synthetic datasets. The frequency-matching framework can be explained as
> $$ \min || \Phi - \hat{\Phi} || = \min || U f(\Lambda) U^{\top} - \hat{U}f(\hat{\Lambda})\hat{U}^{\top}||,$$ where $f$ is a filter and $f(\lambda)$ denotes the filtered eigenvalues.
>
> When applying a low-pass filter, e.g., $f(\lambda)=\lambda$, the low-frequency eigenvectors have larger filtered eigenvalues and therefore dominate the FFC matrix. As a result, the synthetic FFC matrix will first learn low-frequency information to fit the real FFC matrix.
>
> On the contrary, the high-pass filter, e.g., $f(\lambda)=\lambda^{-1}$, will assign larger filtered eigenvalues to high-frequency eigenvectors, thus forcing the synthesis of FFC matrices to learn high-frequency information first.
>
> &nbsp;
>
> > **Q4: Ablation study**
>
> A5: We report the result of the basic classification loss as a baseline in the ablation study.
>
> | | CIFAR | Tiny | ImageNet |
> | --- | --- | --- | --- |
> | $\mathcal{L}_{cls}$ | 55.8 | 46.3 | 40.5 |
> | $\mathcal{L}_{filter}$ | 65.0 | 51.2 | 53.4 |
>
> We can see that the performance of basic classification loss is outperformed by the filter-matching loss, verifying the effectiveness of the frequency-matching framework.
>
> &nbsp;
>
> > **Q5: Results on CIFAR-100**
>
> A6: The ResNet-18 architecture achieves 75.6\% accuracy in the original CIFAR-100 datasets, while CFM achieves 71.4\% accuracy, which is close to the real dataset.

---

### Official Review · Reviewer_71it · 2025-10-29

**Soundness:** 3
**Presentation:** 3
**Contribution:** 3
**Rating:** 6
**Confidence:** 4

**Summary:**

This work explores dataset distillation from a spectral perspective and introduces a new method called Class-wise Frequency Matching (CFM), which aligns the frequency characteristics of real and distilled data. The authors argue that dataset distillation naturally acts like a frequency filter, keeping the important, discriminative details while removing redundant information. Experiments on CIFAR-10 and ImageNet-100 support this idea and show that CFM can improve generalization performance, lending further evidence to their spectral interpretation. To the best of my knowledge, this is the first work that attempts to unify the theoretical understanding and methodological design of dataset distillation under a single spectral framework.

**Strengths:**

1. The paper presents, to the best of my knowledge, the first theoretical framework that systematically analyzes dataset distillation. The spectral analysis connects data condensation with the preservation of high-frequency, discriminative components. The derivations are careful and mathematically consistent, giving a logically sound and interpretable explanation of how synthetic data behave as spectral filters during training.

2. The proposed CFM algorithm makes the theoretical idea practical. CFM matches the class-wise frequency distributions between real and distilled datasets to maintain frequency balance during training (e.g., Eq. 10). This turns the paper’s insight into a concrete algorithm, and the results show steady accuracy gains across different baselines, suggesting that frequency regularization can help dataset distillation generalize better.

3. The writing is clear and the work is easy to reproduce. The mathematical reasoning is well organized, and the authors describe the implementation details for spectral energy computation in sufficient depth. They also release source code for both the analysis and CFM method.

**Weaknesses:**

The experiments are mainly conducted on smaller datasets like CIFAR-10 and ImageNet-100, so it remains uncertain whether the same spectral behavior and CFM improvements would hold on larger datasets such as ImageNet-21K. Adding results on a larger benchmark would make the conclusions more broadly convincing and show the robustness of the proposed approach. In addition, while CFM brings consistent improvements, its performance is still slightly behind the strongest existing methods. As shown in Table 3, the gains are modest and sometimes below strong baselines like RDED or FRePo, suggesting that there is still room for further refinement and optimization of the spectral alignment idea.

**Questions:**

Please check Weaknesses for details.

---

> ### Author Response · Authors · 2025-11-30
> **Response to Reviewer 71it**
>
> We appreciate your detailed review and the recognition of our contributions.
>
> &nbsp;
>
> > **W1: Large-scale dataset**
>
> A1: We kindly clarify that we make experiments on datasets across different scales. The small datasets include CIFAR-10 and CIFAR-100. The large datasets include Tiny-ImageNet (200 classes) and ImageNet-1k (1000 classes). As the reviewer suggested, we also make experiments on the ImageNet-21k datasets. Results are shown below.
>
> | ImageNet-21k | SRe2L | CDA | CFM |
> | --- | --- | --- | --- |
> | IPC-10 | 18.5 | 22.6 | **25.8** |
> | IPC-20 | 20.5 | 26.4 | **27.9** |
>
> We can see that CFM consistently outperforms existing baselines, including SRe2L [1] and CDA [2], indacating its effectiveness in large-scale datasets.
>
> [1] Squeeze, Recover and Relabel: Dataset Condensation at ImageNet Scale From A New Perspective.
>
> [2] Dataset Distillation in Large Data Era.
>
> &nbsp;
>
> > **W2: Performance of CFM**
>
> A2: CFM performs better than FrePo in CIFAR-100, but worse in CIFAR-10. The reason is that FrePo trains multiple distillation networks to prevent the synthetic data from overfitting, while CFM only uses one network. Therefore, CFM has more advantages in the distillation of large datasets.

---

### Official Review · Reviewer_UzEG · 2025-10-31

**Soundness:** 2
**Presentation:** 3
**Contribution:** 2
**Rating:** 4
**Confidence:** 5

**Summary:**

The paper proposed the combination technique, which produce both blurring (low-pass filter) and fine-grained (high-pass filter) image for dataset distillation. Experiments across various benchmarks demonstrate its superior performance over existing baselines.

**Strengths:**

1. The paper is well written and easy to follow.
2. The main idea is clear, and it is easy to understand how it can lead to performance improvement.

**Weaknesses:**

1. The idea of using both blurring and fine-grained techniques for DD is not novel, as it has already been explored in [1].
2. The proposed method only works under high IPC settings (greater than 1 IPC).
3. The paper lacks comparisons with several state-of-the-art methods, such as [1] and [2].

[1] Enhancing Dataset Distillation via Non-Critical Region Refinement. CVPR 2025
[2] Dataset Distillation via the Wasserstein Metric. ICCV 2025

**Questions:**

See weaknesses section!

---

> ### Author Response · Authors · 2025-11-30
> **Response to Reviewer UzEG**
>
> We are grateful for your constructive advice and the opportunity to address your concerns.
>
> &nbsp;
>
> > **W1: Difference between UniDD and NRR-DD**
>
> A1: The methodology and motivation between UniDD and NRR-DD are different.
>
> - NRR-DD considers the blurring and fine-grained features in the **spatial domain**, while UniDD leverages the low-frequency and high-frequency information in the **spectral domain**.
> - NRR-DD uses a pixel-wise mask to extract the **fine-grained objectives**, while UniDD aims to seek the **high-frequency eigenvectors** of the image representations.
> - The motivation behind UniDD is to reveal the connections between various DD methods, rather than to combine low-frequency and high-frequency information.
>
> In summary, although UniDD and NRR-DD both use the blurring and fine-grained features, the two methods are different in both methodology and motivation.
>
> &nbsp;
>
> > **W2: Results on 1-IPC**
>
> A2: We make an additional experiment on the extremely low budget, i.e., one-image-per-class (1-IPC). The results are shown below. We can see that CFM outperforms baselines in the low IPC setting.
>
> | | CIFAR-10 | CIFAR-100 | Tiny | ImageNet-1k |
> | --- | --- | --- | --- | --- |
> | SRe2L | 16.6 | 6.6 | 2.6 | 0.1 |
> | RDED | 22.9 | 11.0 | 9.7 | 6.6 |
> | CFM | **25.8** | **18.9** | **10.5** | **7.0** |
>
> &nbsp;
>
> > **W3: Comparision with SOTA methods**
>
> A3: As suggested by Reviewer UzEG, we add two new baselines: NRR-DD[1] and WMDD [2] in the Tiny-ImageNet and ImageNet-1k datasets under 10-image-per-class (10-IPC).
>
> | 10-IPC | Tiny | ImageNet |
> | --- | --- | --- |
> | NRR-DD | 45.2 | 46.1 |
> | WMDD | 41.8 | 38.2 |
> | CFM | 43.3 | 40.6 |
>
> We can observe that CFM can outperform WMDD, but is weaker than NRR-DD. A possible reason is that NRR-DD uses RDED as initialization, while CFM uses the raw images.

---

### Official Review · Reviewer_zry6 · 2025-10-31

**Soundness:** 3
**Presentation:** 3
**Contribution:** 3
**Rating:** 4
**Confidence:** 4

**Summary:**

This study points out that existing dataset distillation objective functions have developed in a fragmented manner, and their relationships have not been investigated. This study proposes a new framework, UniDD, utilizing spectral filtering. Furthermore, UniDD enables to categorize previous researches based on the frequency bands the filter function focuses upon. Additionally, this work proposes Curriculum frequency matching (CFM), enabling the learning of both low and high-frequency information, thereby mitigating the limitations of prior research that only learned information from specific frequency bands.

**Strengths:**

1. It is both highly impressive and fascinating that reinterpreting the objective function of existing dataset distillation from a spectral filtering perspective and proposing a unified framework.

2. The proposed UniDD is a simple form, and the logical flow of expressing prior research using UniDD is also readily comprehensible.

3. The motivation and core idea of CFM, which adaptively learns multiple frequency information based on UniDD, is natural and intuitive.

**Weaknesses:**

1. UniDD was proposed under assumptions that are prone to violation. Overall, this work develops UniDD’s logic under the assumption that the parameters of the feature extractor $\phi$ are fixed and not considered. Consequently, both gradient matching and trajectory matching only considered the parameters of a single layer following the feature extractor. However, in dataset distillation, the parameters of the feature extractor are typically not fixed, meaning the UniDD framework may not readily satisfy these conditions. This suggests that UniDD has limitations in terms of its applicability.

2. The objective functions for gradient matching and trajectory matching represent lower bounds for UniDD. This implies that achieving gradient matching and trajectory matching can be accomplished through low-frequency matching and high-frequency matching respectively; however, it is difficult to accept that the converse holds true. Therefore, the claim that gradient matching and trajectory matching correspond to low-frequency matching and high-frequency matching requires further clarification.

3. The theoretical analysis and experimental design in this study do not align well. UniDD integrates statistical matching, gradient matching, trajectory matching, and KRR from a spectral filtering perspective, and proposes CFM to learn diverse frequency information. However, the baseline used in all experimental results except Table 2 is the decoupled method, which is difficult to consider as included in UniDD. To demonstrate the superiority of CFM, I believe that it is necessary to conduct diverse performance comparison experiments against the recent works for each component included within UniDD.

4. Key previous literature is omitted. FreD[1], NSD[2], and NCFM[3] are prior studies addressing the frequency domain in dataset distillation; therefore, the relevance of the proposed idea must be discussed in the related works section.

[1] Frequency Domain-based Dataset Distillation

[2] Neural Spectral Decomposition for Dataset Distillation

[3] Dataset Distillation with Neural Characteristic Function: A Minmax Perspective

**Questions:**

1. To support the claim that DM and DC are low-frequency matching methods while MTT and KRR are high-frequency matching methods, I suggest plotting the spectral density for the synthetic images generated by each methodology.

2. I believe the synthetic label $Y_s$ need not necessarily be a one-hot label; treating it as an optimizable parameter does not alter the overall logic. Furthermore, in the experiments, soft labels based on fast knowledge distillation (FKD) were employed. I would be interested whether CFM remains effective when using optimized synthetic soft-labels.

3. When using soft labels as synthetic labels, more memory is required than when storing only synthetic inputs, as the labels themselves must also be saved. Furthermore, the memory usage of synthetic labels increases significantly as the number of classes grows, to a point where it becomes non-negligible[4]. Therefore, I am curious how the utilized memory budget was configured to ensure a fair comparison with the baseline.

4. I am interested in ablation studies concerning various frequency curriculum scheduling approaches. I am also curious whether this can be applied to parameterization methodologies such as FreD[1] and NSD[2], which only involve fixed frequency components in learning.

5. I would be interested whether CFM exhibits high performance even when IPC=1.

[4] Are Large-scale Soft Labels Necessary for Large-scale Dataset Distillation?

---

> ### Author Response · Authors · 2025-11-30
> **Response to Reviewer zry6 (1/2)**
>
> Thanks for the detailed and helpful comments. We reply to the comments in detail. Hope to address your concerns.
>
> &nbsp;
>
> > **W1: Assumptions in UniDD**
>
> A1: A1: Our theoretical framework is based on the analysis of a linear classifier. We think existing results have provided enough insights about dataset distillation. The reasons are two-fold:
>
> - Many existing methods employ fixed encoders during the distillation process. For example, FrePo [1] uses fixed image encoders and models the linear classifier as a closed-form regression loss, which has achieved considerable results. Moreover, SRe2L [2] also uses a fixed ResNet-18 as the distillation backbone and extends DD to ImageNet-1k successfully.
> - Although gradient matching and trajectory matching use full parameters, they do not show competitive results on large-scale datasets, e.g., ImageNet-1k. Moreover, some recent works [3, 4] show that matching gradients or trajectories of the last layer can obtain better distillation performance, which supports our theoretical analysis.
>
> [1] Dataset Distillation using Neural Feature Regression.
>
> [2] Squeeze, Recover and Relabel: Dataset Condensation at ImageNet Scale From A New Perspective.
>
> [3] Dataset Distillation for Pre-Trained Self-Supervised Vision Models.
>
> [4] Beyond Modality Collapse: Representations Blending for Multimodal Dataset Distillation.
>
> &nbsp;
>
> > **W2 & Q1: DC and MTT for low-frequency matching and high-frequency matching**
>
> A2: We thank the reviewer for pointing out the spectral density experiment to verify the conclusion of UniDD. Below, we report the average value of the radial power spectral density (RPSD) of DC and MTT to show their differences in the spectral domain.
>
> Specifically, we calculate the synthetic images learned by DC and MTT in the CIFAR-10 and CIFAR-100 datasets. Each image has a resolution of 32x32. Therefore, the radius is $23 > \sqrt{16^2 + 16^2} \approx 22.62$.
>
>
> ``CIFAR-10``
>
> DC: [1.5e-09, 6.6e+04, 1.4e+04, 5.7e+03, 2.9e+03, 1.4e+03,
>   8.9e+02, 5.7e+02, 3.9e+02, 2.7e+02, 1.9e+02, 1.5e+02,
>   1.0e+02, 7.4e+01, 5.7e+01, 4.4e+01, 3.1e+01, 1.1e+01,
>   7.6e+00, 5.2e+00, 4.1e+00, 3.0e+00, 2.5e+00]
>
> MTT: [6.5e-09, 1.6e+05, 1.3e+05, 5.5e+04, 4.6e+04, 2.7e+04,
>   1.0e+04, 8.0e+03, 7.2e+03, 5.4e+03, 3.7e+03, 3.3e+03,
>   2.4e+03, 2.2e+03, 1.9e+03, 1.8e+03, 1.5e+03, 9.6e+02,
>   8.5e+02, 8.6e+02, 8.6e+02, 9.1e+02, 8.0e+02]
>
> `` CIFAR-100``
>
> DC: [2.8e-10, 7.2e+04, 8.5e+03, 4.2e+03, 3.6e+03, 2.1e+03,
>   1.1e+03, 1.1e+03, 9.3e+02, 5.6e+02, 4.1e+02, 3.4e+02,
>   3.0e+02, 2.8e+02, 3.3e+02, 3.4e+02, 2.4e+02, 2.1e+02,
>   2.0e+02, 1.9e+02, 2.0e+02, 1.9e+02, 1.6e+02]
>
> MTT: [2.2e-08, 2.8e+05, 1.1e+05, 6.6e+04, 5.5e+04, 2.5e+04,
>   1.3e+04, 1.0e+04, 9.4e+03, 6.1e+03, 4.5e+03, 4.0e+03,
>   3.4e+03, 2.6e+03, 2.5e+03, 2.4e+03, 1.7e+03, 1.0e+03,
>   9.6e+02, 9.4e+02, 9.3e+02, 9.6e+02, 1.0e+03]
>
> We can observe that the RPSD of MTT consistently outperforms those of DC, validating that MTT is high-frequency matching and DC is low-frequency matching.
>
> &nbsp;
>
> > **W3: Decouple method for CFM**
>
> A3: The goal of UniDD is to reveal the connections between various DD methods. We have included a series of representative works, e.g., DC, DM, MTT, and KRR. However, we do not claim that UniDD can unify all DD methods.
> On the other hand, these representative works are hard to generalize to large-scale datasets, e.g., ImageNet-1k. Therefore, we extend CFM into the decouple framework to ensure that it can achieve considerable results in the large-scale datasets. The results in Tables 2 and 3 can validate the superiority of CFM in both small and large datasets.
>
> &nbsp;
>
> > **W4: Compared to other frequency domain DD methods**
>
> A4: Some existing works, e.g., FreD [5], leverage the Fourier transform to convert images to the frequency domain and perform dataset distillation. While effective, these works differ from UniDD in the following perspectives.
>
> - UniDD studies the **dataset-level frequency**, i.e., $X^{T}X$, where $X \in \mathbb{R}^{n \times d}$. In contrast, FreD focuses on **sample-level frequency**, i.e., a single image, which leverages the DCT to parametrize each synthetic image.
> - UniDD mainly analyzes the **objective functions** of various DD methods and directly optimizes the synthetic images in the pixel space. On the contrary, FreD is a **data parameterization** method, which updates the synthetic images in the spectral domain. It can be combined with an arbitrary DD objective.
> - In summary, UniDD and FreD are two orthogonal research lines that can promote each other. We will cite this work and add this discussion in the revision.
>
> [5] Frequency Domain-based Dataset Distillation.

---

> ### Author Response · Authors · 2025-11-30
> **Response to Reviewer zry6 (2/2)**
>
> > **Q2 & Q3: Soft-labels used in CFM**
>
> A5: We discuss two ways to use soft labels in the pipeline of CFM.
>
> - Training with soft labels. Replacing the one-hot labels $Y_s$ with the corresponding soft labels does not affect the derivation of UniDD. However, during experiments, we find that the performance of one-hot labels and soft labels is similar. Therefore, we do not use the soft labels in the distillation process.
> - Evaluation with soft labels. Using knowledge distillation to improve the performance of DD is a widely used trick in the large-scale datasets, e.g., Tiny-Imagenet and Imagenet-1k. We strictly follow the settings of SRe2L to ensure that all baseline models have the same memory budget.
>
> &nbsp;
>
> > **Q4: Curriculum scheduling and applications to other DD methods**
>
> A6: We compare the performance of cosine scheduling, linear scheduling, and fixed frequency filters. We can see that the cosine and linear scheduling have similar performance, and they consistently outperform the fixed frequency filters, validating the effectiveness of CFM.
>
> | IPC=50 | Low-pass | High-pass($\beta=0.1$) | CFM (cosine) | CFM (linear)|
> | --- | --- | --- | --- |  --- |
> | Tiny | 53.1 | 53.9 | 54.8 | 54.7 |
> | ImageNet | 55.5 | 55.8 | 56.9 | 56.9 |
>
> As discussed in the Related Work section, UniDD focuses on analyzing the **objective functions** of various DD methods, which is orthogonal to the parameterization-based DD methods. Therefore, UniDD can be applied to other frequency domain DD methods, such as FreD.
>
> &nbsp;
>
> > **Q5: Results on 1-IPC**
>
> A7: We make an additional experiment on the extremely low budget, i.e., one-image-per-class (1-IPC). The results are shown below. We can see that CFM outperforms baselines in the low IPC setting.
>
> | | CIFAR-10 | CIFAR-100 | Tiny | ImageNet-1k |
> | --- | --- | --- | --- | --- |
> | SRe2L | 16.6 | 6.6 | 2.6 | 0.1 |
> | RDED | 22.9 | 11.0 | 9.7 | 6.6 |
> | CFM | **25.8** | **18.9** | **10.5** | **7.0** |

---

### Meta-Review · Area_Chair_oiq9 · 2025-12-22

**Summary:**

Some major concerns are shared across reviewers:
1. The theoretical analysis and experimental design are not well aligned.
2. Illustration of high-frequency matching and low-frequency matching.
3. Reviewers asked for experiments on larger datasets, 1-IPC setting, and comparison with recent state-of-the-art methods.
4. The lack of comparison and discussion with image-level frequency matching.
5. Extra compute is required for the proposed Curriculum Frequency Matching.

**Reviewer Concerns:**

In the rebuttal, the authors provided detailed justification of their experimental design and requested experiment results. They also proved that the proposed method only introduced marginal additional compute.

**Reviewer Scores:**

The concerns were largely addressed by the rebuttal. I think the reviewers would agree to accept the paper.

---

### Decision · Program_Chairs · 2026-01-26

Accept (Poster)